# A Spatiotemporal Atlas of the Gut Microbiota in *Macaca mulatta brevicaudus*: Implications for Health and Environment

**DOI:** 10.3390/biology14080980

**Published:** 2025-08-01

**Authors:** Jingli Yuan, Zewen Sun, Ruiping Sun, Jun Wang, Chengfeng Wu, Baozhen Liu, Xinyuan Zhao, Qiang Li, Jianguo Zhao, Keqi Cai

**Affiliations:** 1Innovation Center of Academician Xia Xianzhu’s Team, Key Laboratory of Tropical Animal Breeding and Disease Research, Institute of Animal Science and Veterinary Medicine, Hainan Academy of Agricultural Sciences, Haikou 571100, China; 13250732023@163.com (J.Y.); ruiping937@126.com (R.S.); junw99@outlook.com (J.W.); 18234704422@163.com (Q.L.); 2Sanya Institute, Hainan Academy of Agricultural Sciences, Sanya 572019, China; 3State Key Laboratory of Stem Cell and Reproductive Biology, Institute of Zoology, Stem Cell and Regenerative Medicine Innovation Institute, Chinese Academy of Sciences, Beichen West Road, Chaoyang District, Beijing 100101, China; sunzewen@ioz.ac.cn; 4College of Veterinary Medicine, Xinjiang Agricultural University, Urumqi 830052, China; 5School of Life Sciences, Sun Yat-sen University, Guangzhou 510275, China; wuchf27@mail.sysu.edu.cn; 6College of Veterinary Medicine, South China Agricultural University, Guangzhou 510642, China; bushiaikugui@126.com (B.L.); 18893488214@163.com (X.Z.); 7Tropical Crop Genetic Resource Research Institute, Chinese Academy of Tropical Agricultural Sciences, Haikou 571101, China

**Keywords:** *Macaca mulatta brevicaudus*, metagenome, age, sex, physiological state

## Abstract

This study employed metagenomic sequencing to analyze differences in the gut microbiota of *M. mulatta brevicaudus* across age, sex, and physiological states. We identified distinct dominant bacterial species (e.g., *Segatella copri*) and metabolic functions (e.g., carbohydrate metabolism) among different groups (e.g., infant/young/middle/old, male/female, lactating/non-lactating). These findings not only provide new insights into host–microbiota coevolution in primates but also establish a foundation for developing microbiome-based therapies for human intestinal disorders, such as metabolic or immune-related diseases.

## 1. Background

*Macaca mulatta* is the most widely distributed non-human primate species, with diverse habitats that make it an ideal model for studying the adaptive evolution of non-human primates [1]. Given its significant physiological similarities to humans, the macaque has been extensively utilized in research related to human diseases [2]. Research indicates that the composition and diversity of the gut microbiota in *Macaca mulatta* are influenced by various factors, including age, sex, physiological state, dietary habits, and environmental conditions [3,4]. These studies suggest a close relationship between the structure and functionality of the gut microbiota and the health status of the host. Dysbiosis of the gut microbiota may contribute to metabolic diseases such as diabetes, obesity, and immune-related conditions (e.g., inflammatory bowel disease) [5,6,7] as well as neurological disorders (e.g., Alzheimer’s disease, autism) [8,9,10]. The pathogenic roles of gut microbiota in obesity and diabetes have also been extensively investigated for therapeutic drug development [11]. Recent studies suggest that gut microbiota may influence the onset of diseases such as Alzheimer’s disease and autism through the gut–brain axis [12]. Moreover, dysbiosis in the gut microbiota of non-human primates may affect reproductive success, growth, development, and disease resistance [13]. Previous studies have revealed that the gut microbiota of *Macaca mulatta* is predominantly composed of bacterial phyla typical in primates, including Firmicutes, Bacteroidetes, and Spirochaetes [14,15]. As a keystone species in tropical forest ecosystems, the structural characteristics of its gut microbiota not only reflect the physiological impacts of environmental stressors but also provide novel microbial-level evidence for assessing adaptive evolution under habitat fragmentation. These findings establish a theoretical foundation for developing conservation strategies for endangered primates [16,17].

*M. mulatta brevicaudus*, a distinct member of the macaque family, is not only one of the most important distribution types of macaques in China but also a significant subject for research on adaptive evolution, ecological conservation, and disease prevention and control. This subspecies is primarily distributed on Hainan Island, where its unique ecological environment and genetic background provide valuable natural resources for studies involving model organisms [18]. As microbiome research deepens, the role of gut microbiota in animal health, disease, and nutritional metabolism has garnered increasing attention. The unique geographic isolation and long-term adaptation of *M. mulatta brevicaudus* to tropical environments have resulted in specialized physiological and behavioral traits, with its gut microbiota reflecting dietary resources and environmental adaptability [19]. Dietary diversity may have profound effects on the diversity and functionality of its gut microbiota [20]. Although previous studies have reported gut microbiota analyses of *M. mulatta brevicaudus* and elucidated its microbial diversity, these investigations were primarily conducted for comparative purposes with other species, rather than as comprehensive standalone characterizations [21]. Currently, systematic studies on the gut microbial diversity of macaques across different ages, sexes, and physiological states remain insufficient, particularly regarding *M. mulatta brevicaudus*, which limits our comprehensive understanding of its gut microbial characteristics. Therefore, elucidating the composition and diversity of gut microbiota in *M. mulatta brevicaudus* under various conditions can provide scientific evidence for the conservation and management of this species, as well as for research on disease prevention and ecological adaptability.

Age is one of the significant factors influencing gut microbial diversity in macaques [22,23,24]. Research indicates that the diversity of gut microbiota in macaques is relatively low during infancy and gradually increases with age, resulting in a more complex and stable microbial community structure in adult macaques [25,26,27]. Sex differences are also potential factors influencing the composition and diversity of gut microbiota in macaques [22,24,28]. Although research in this area is relatively limited, some studies suggest that sex may influence gut microbiota balance through hormonal regulation [22]. The impact of different physiological states on gut microbiota in macaques should not be overlooked, particularly during lactation compared to non-lactation periods [29,30]. Lactating females may exhibit differences in gut microbial diversity compared to non-lactating macaques due to changes in physiological needs and nutritional metabolism, reflecting physiological adjustments made to meet lactation demands, including alterations in energy metabolism, nutrient absorption, and immune function [31,32,33].

Therefore, this study aims to conduct a systematic analysis of gut microbial diversity in *M. mulatta brevicaudus* across different ages, sexes, and physiological states (lactation versus non-lactation) using metagenomic sequencing technology. This analysis will include patterns of microbial variation, community characteristics, and their functional impacts, thereby elucidating the mechanisms of interaction between the host and microorganisms. This research will help fill the gap in understanding the gut microbial community characteristics of the Hainan subspecies under various conditions, further enriching the comprehensive understanding of the gut microbiota diversity in macaques. It will also provide valuable perspectives and insights for research on the relationship between gut microbiota and health or disease.

## 2. Materials and Methods

### 2.1. Experimental Animals and Sample Collection

In this study, 41 fecal samples from *M. mulatta brevicaudus* were collected from eight breeding groups in the Nanwan Nature Reserve in Hainan, encompassing various ages, sexes, and lactation statuses. The nature reserve is located on a peninsula surrounded by sea on three sides in Hainan Island (109°59′ E, 18°24′ N), covering approximately 10 km^2^ of a tropical monsoon climate region. The peninsula extends east–west, with a maximum elevation of 255 m and vegetation dominated by secondary evergreen monsoon forest. The total population of macaques on Nanwan Monkey Island is approximately 1900 individuals. The reserve maintains designated artificial feeding areas providing supplementary foods including rice, peanuts, and fruits, while the macaques also forage naturally on native vegetation, fruits, and insects within their wild habitat. We sampled four different age groups of macaques: infant, young, middle-aged, and old, as detailed in Table 1. Each sample was collected within two minutes after defecation to ensure timely collection into sterile containers, and the samples were subsequently stored in liquid nitrogen. This study prioritized the welfare of the wildlife involved and did not employ any methods that could potentially harm the subjects. The research adhered to relevant laws and regulations, including China’s Wildlife Protection Law and the Management Regulations for Nature Reserves.

### 2.2. DNA Extraction and Detection

DNA was extracted from fecal samples using the Magnetic Bead Fecal and Soil Genome Extraction Kit (MAGEN, Guangzhou, China) following the manufacturer’s instructions (https://www.yuque.com/yangyulan-ayaeq/oupzan/krtheyyvbu9au4lo, accessed on 29 July 2025). The purity and concentration of the extracted DNA were assessed using 1% agarose gel electrophoresis to ensure that the DNA was suitable for amplification.

### 2.3. Metagenomic Sequencing

Library preparation was performed by using the MGIEasy Universal DNA Library Prep Set (MGI, Shenzhen, China). A specific amount of genomic DNA is taken and subjected to fragmentation processing. Fragmented samples are size-selected through magnetic beads, yielding library inserts concentrated in the 200–400 bp range. Then, the size-selected DNA fragments are converted to blunt-end DNA with an end repair reaction. A single Adenosine nucleotide is added on the 3’ end of DNA through the A-tailing reaction. Later, the library adapters are connected to the two ends of the DNA by adaptor ligation. Finally, the library products are amplified through PCR reaction and subjected to a quality control process.

### 2.4. Data Analysis

The raw data were subjected to quality filtering using the SOAPnuke (v2.3) software. Bowtie2 (2.4.4) was then employed to align host sequences, removing any matching sequences to generate clean data. The quality-controlled clean data underwent de novo assembly using the MEGAHIT (1.2.9) software [34] filtering out assembled sequences shorter than 300 bp, followed by statistical analysis and subsequent gene prediction. The assembly results were evaluated based on continuity (Assembly Length, Max Length) and completeness (N50, N90, Min Length) as assessment criteria. Gene prediction for the metagenome was performed using MetaGeneMark (3.38) [35], followed by the deduplication of gene prediction results for each sample using the CD-HIT (4.8.1) software [36]. Subsequently, gene quantification was carried out using the Salmon (1.6.0) software [37], yielding standardized gene abundance values.

Species annotation was performed using Kraken2 (2.1.2) with default parameters, and the species-level abundance of metagenomic samples was estimated through Bracken2 (2.6.1) using the Bayesian algorithm along with Kraken classification results. Inter-group differences were assessed by calculating the Euclidean distance, Bray–Curtis distance, and Jensen–Shannon divergence, which reflect Beta diversity [38] and indicate whether there is a significant difference in microbial community composition among groups. Non-parametric statistical tests (Wilcoxon/Kruskal–Wallis) were performed on the abundance of all species, with *p* < 0.05 considered statistically significant for identifying differentially abundant species between groups. Functional annotation of non-redundant genes was conducted using the BLASTP functionality of the Diamond (0.8.24) software [39], aligning the results with the KEGG (Kyoto Encyclopedia of Genes and Genomes; version: 101) and CARD (The Comprehensive Antibiotic Resistance Database; version: 3.0.9) databases.

The random forest model is a powerful tool in microbial research, effectively selecting and identifying important microorganisms, thereby aiding in the understanding of the complexity and functionality of microbial communities [40]. We identified potential biomarker species through analytical approaches and subsequently validated their predictive accuracy for environmental changes. The methodology primarily involved random forest analysis to select key microbial biomarkers and construct an environmental change prediction model, and ROC analysis to evaluate the diagnostic accuracy of the developed model. The R (4.3.1) package “randomForest” was employed for the selection of important metabolic features in this study.

## 3. Results

### 3.1. Sequencing and Observation

Metagenomic sequencing was conducted on the fecal samples from 41 macaques. The assembly outcomes were evaluated based on continuity (Assembly Length, Max Length) and completeness (N50, N90, Min Length). According to Table A1, the average length was 1402.73 bp. Following quality control, assembly, and prediction, gene information from the metagenome was obtained, with gene lengths primarily distributed between 200 and 499 nt (Figure A1). Species accumulation curves, reflecting species richness with increasing sampling effort, showed that dilution curves for different macaque groups plateaued, indicating sufficient sampling volume (Figure 1A–C). Based on years of longitudinal observations, we captured images of the same monkey at different age stages (Figure 1D–G).

### 3.2. Microbial Diversity Across Different Age Groups of Macaques

To analyze the diversity of fecal microbial communities in macaques of different age groups, we assessed the Beta diversity of microorganisms in each sample. As shown in Figure 2A, there were significant differences in microbial composition at the species level between the juvenile and middle-aged groups (*p* < 0.05), as well as between the young adult and middle-aged groups (*p* < 0.05). The ANOSIM analysis revealed statistically significant differences (*R* = 0.0969, *p* = 0.036, *R* denotes the ANOSIM test statistic, representing the degree of separation between predefined groups (range: −1 to 1)). The results of Partial Least Squares Discriminant Analysis (PLS-DA) indicated that the microbial composition at the species level exhibited both similarities and differences across different age groups, with notable distinctions between the young adult and middle-aged groups, while the juvenile and elderly groups showed higher similarity (Figure 2B), and at the phylum and genus levels demonstrated similar patterns to those observed at the species level (Figure A2C,D).

To understand the abundance of fecal microorganisms in macaques at different life stages, we analyzed the species-level abundance of fecal microbiota across various age groups (Figure 2D). The microbial species *Segatella copri* was consistently the most abundant across all age groups, followed by *Segatella hominis* and *Faecalibacterium prausnitzii*, with *Segatella copri* exhibiting higher abundance during the juvenile stage compared to other periods. At the genus level, we found that the genus *Segatella* was the most abundant across different age groups, with significantly higher abundance during the juvenile stage as well (Figure A3B). At the phylum and the genus level, the most abundant microbial group in the feces of macaques across all age groups was *Bacillota* (Figure A3A). Differential analysis at the species level revealed that *Bifidobacterium adolescentis* had significantly higher abundance during the juvenile stage compared to other periods (*p* < 0.05), which may be closely related to its role during the growth phase of juveniles (Figure 2E). Other key differential microbes in *M. mulatta brevicaudus* across different life stages included *Desulfovibrio piger*, *Wujia chipingensis,* and *Faecalibacterium sp. I4-3-84*. To assess the shared and unique species across different life stages, we analyzed Venn diagrams at the species level for various age groups (Figure 2C). A total of 2747 microbial species were found to be common across the different periods, with the middle-aged group exhibiting the highest number of unique species (700 species). At the genus level (Figure A4B), there were 1334 shared microbial genera, with the middle-aged group again having the largest number of unique genera (116 species). At the phylum level (Figure A4A), there were 53 shared microbial phyla across the different age groups. Through the screening of significant microbial differences among various age groups (Figure 2F), key microorganisms identified include *Burkholderia stabilis*, *Coprococcus comes*, *Bacteroides thetaiotaomicron,* and *Bifidobacterium adolescentis*.

### 3.3. Microbial Diversity in Male and Female Macaques

To understand the diversity of fecal microbial communities in male and female macaques, we assessed the Beta diversity of microorganisms in each sample. Our analysis revealed no significant differences in the fecal microbiota between female and male macaques at any level (*p* < 0.05) (Figure 3A, Figure A5A,B). The ANOSIM analysis revealedno statistically significant differences (*R* = −0.0271, *p* = 0.58). The results of the PLS-DA analysis (Figure 3B, Figure A5C,D) indicated that the microbial composition at the phylum, genus, and species levels demonstrated considerable similarity between the sexes. Differences in the composition and functionality of the gut microbiome between male and female macaques may impact their overall health, immune function, and behavioral performance. To evaluate the abundance of fecal microorganisms in female and male macaques, we analyzed the species-level abundance of fecal microbiota (Figure 3D). The microbial species *Segatella copri* was found to be the most abundant in both sexes, followed by *Segatella hominis*. At the genus level, *Segatella* was again the most abundant genus among both males and females, with *Faecalibacterium* following closely (Figure A6B). At the phylum level, the most abundant microbial phylum in the macaque feces was *Bacillota*, consistent across both sexes (Figure A6A). In the differential analysis at the species level (Figure 3F), it was observed that the abundances of *Alistipes senegalensis*, *Parabacteroides merdae*, *Parabacteroides johnsonii* and *Parabacteroides sp. CT06* were significantly lower in male macaques compared to females (*p* < 0.05). At the genus level (Figure A6D), *Vibrio, Pseudoalteromonas, Arsenophonus,* and *Actinobacillus* were found to have significantly lower abundances in male macaques (*p* < 0.05), with *Arsenophonus* showing particularly high abundance in female macaques, suggesting its potential key role in their microbiome. At the phylum level, *Artverviricota* was the only microbial group with significant differences observed between the sexes (*p* < 0.05). The Venn diagram analysis at the species level revealed that there are 4039 microbial species shared between male and female macaques, with 397 species unique to males and 406 species unique to females (Figure 3C). At the genus level (Figure A7B), a total of 1601 microbial genera are shared between the sexes. At the phylum level (Figure A7A), there are 56 shared microbial phyla. The Linear Discriminant Analysis Effect Size (LEfSe) method can be utilized to identify species that best explain the differences between two groups (known as biomarkers) and to assess the extent of their impact on intergroup differences through Linear Discriminant Analysis (LDA), making it suitable for multi-level discovery and interpretation. The results of the LEfSe analysis for different sexes identified the species *Alistipes senegalensis*, *Parabacteroides johnsonii*, *Parabacteroides merdae*, *Parabacteroides sp. CT06,* and *Aeromonadales* as the most significant in explaining the differences between males and females (Figure 3H). The LDA values (LDA = 2) indicated (Figure 3I) that *Alistipes senegalensis and Aeromonadales* had the greatest influence among the species between the sexes. Through the random forest model, the AUC value was 0.713 (Figure 3E), and the model prediction reliability was high; the important microorganisms were *Succinivibrio dextrinosolvens*, *Treponema succinifaciens*, *Ligilactobacillus ruminis*, *Klebsiella pneumoniae,* and *Bifidobacterium adolescentis* (Figure 3G).

### 3.4. Microbial Diversity During Lactation and Non-Lactation Periods in Macaques

Analysis of the fecal microbial communities in macaques during lactation and non-lactation periods revealed significant differences at the species level (*p* < 0.05) (Figure 4A). The ANOSIM analysis revealed no statistically significant differences (*R* = −0.0988. *p* = 0.879). Similarly, significant differences were observed at both the genus and phylum levels between lactation and non-lactation periods (*p* < 0.05) (Figure A8A,B). The results from the PLS-DA analysis (Figure 4B) demonstrated distinct differences in the microbial composition at the species level between these two states. Additionally, PLS-DA analyses at the genus and phylum levels showed similar results, indicating clear differences as well (Figure A8C,D). The differences in gut microbiota between lactating and non-lactating macaques may be a crucial factor influencing the health of juvenile macaques’ microbiomes. Therefore, understanding the abundance of fecal microorganisms during these two periods is essential. The species-level abundance analysis revealed that *Segatella copri* is the dominant microbial species in both lactating and non-lactating macaques, followed by *Faecalibacterium prausnitzii* and *Segatella hominis* (Figure 4D). At the genus level, *Segatella* remains the most abundant in both periods, with *Faecalibacterium* and *Vescimonas* also being prominent (Figure A9B). At the phylum level, *Bacillota* and *Bacteroidota* were the most abundant in macaque feces during both lactation and non-lactation periods (Figure A9A). In the differential analysis of microbial species between lactating and non-lactating macaques (Figure 4F), it was observed that *Alistipes indistinctus*, *Sulfurivermis fontis,* and *Brachyspira pilosicoli* had significantly lower abundances during lactation compared to the non-lactation period (*p* < 0.05). Conversely, *Streptomyces sp. T12* and *Capnocytophaga haemolytica* exhibited significantly higher abundances during lactation. At the genus level (Figure A9C), the abundances of *Sulfurivermis, Kushneria*, and *Malaciobacter* in lactating macaques were significantly lower than in non-lactating macaques (*p* < 0.05). The Venn diagram analysis at the species level revealed that there are 4026 microbial species shared between lactating and non-lactating macaques, with 360 species unique to the lactation period and 715 species unique to the non-lactation period (Figure 4C). At the genus level (Figure A10B), a total of 1572 microbial genera are shared between the two periods. At the phylum level (Figure A10A), there are 53 shared microbial phyla. The results of the LEfSe analysis for the lactation and non-lactation periods (Figure 4H) identified Alistipes indistinctus as the species that best explains the differences between these two states. The LDA value (LDA = 2) indicated (Figure 4I) that Alistipes indistinctus also had the most significant impact between the lactation and non-lactation periods. Through the random forest model screening between lactating and non-lactating microorganisms, the AUC value was 0.688 (Figure 4E), and the model prediction reliability was high; the important microorganisms were *Ligilactobacillus animalis*, *Escherichia coli*, *Butyrivibrio crossotus*, *Blautia luti,* and *Blautia wexlerae* (Figure 4G).

### 3.5. Functional Analysis

Functional annotation of metagenomes involves assigning predicted non-redundant genes to relevant functional databases and quantifying their abundances. To understand the functional distribution of gut microbiota in macaques, we conducted functional annotation of their gut microbiomes.

The KEGG functional enrichment analysis across different age groups revealed (Figure 5A) that, similar to the findings related to sex, the primary enrichment was observed in pathways such as global and overview maps, carbohydrate metabolism, amino acid metabolism, and metabolism of cofactors and vitamins. Notably, the number of genes in middle-aged macaques of the *M. mulatta brevicaudus* was higher than in other age groups. The annotation results for resistance genes (Figure 5B) indicated a predominance of tetracycline antibiotic, Penam, cephalosporin, and aminoglycoside antibiotic. Across the major enriched pathways, middle-aged macaques also exhibited the highest number of genes related to these resistance mechanisms.

The KEGG functional enrichment analysis revealed (Figure 6A) that pathways were primarily enriched in areas such as global and overview maps, carbohydrate metabolism, amino acid metabolism, and metabolism of cofactors and vitamins. In these significantly enriched pathways, the number of genes in males was consistently higher than in females. With the discovery and use of antibiotic drugs, an increasing number of antibiotic-resistant strains have emerged. Identifying resistance genes is crucial for understanding resistance mechanisms, which can inform disease treatment and drug development. Therefore, we utilized the Comprehensive Antibiotic Resistance Database (CARD) to annotate the metagenome of macaque fecal microbiota (Figure 6B). This analysis identified 36 classes of resistance genes, predominantly including tetracycline antibiotic, Penam, cephalosporin, and aminoglycoside antibiotic. Similarly, we found that the abundance of antibiotic resistance genes in males was higher than in females across the major enriched resistance pathways.

The KEGG functional enrichment analysis between lactation and non-lactation periods revealed (Figure 7A) that, similar to the findings related to age and sex, the primary enrichment was observed in pathways such as global and overview maps, carbohydrate metabolism, amino acid metabolism, and metabolism of cofactors and vitamins. The annotation results for resistance genes (Figure 7B) indicated a predominance of tetracycline antibiotic, Penam, cephalosporin, and aminoglycoside antibiotic. In all major enriched pathways, the number of genes during the lactation period was higher than that during the non-lactation period.

## 4. Discussion

Due to their genetic diversity and evolutionary similarities to humans, macaques are often considered important model animals for elucidating human physiological functions, studying clinical diseases, and developing new drugs [41,42]. A healthy gut, as a critical center for signal transduction and metabolic homeostasis, is a key determinant of lifespan [43]. Microbial genomics can provide better insights into the roles of gut microbiota in host metabolism, nutrition, and disease [44,45]. Metagenomics allows for the exploration of microbial species classification, community structure, gene functions, and metabolic networks in the fecal microbiota of macaques [46,47]. Statistical analysis methods can be employed to compare differences in species composition and community functions across different samples, significantly expanding the quantity and diversity of microbial genomes, particularly for unculturable strains [48]. This study investigates the effects of age, sex, and physiological states on the microbiota of *M. mulatta brevicaudus*, providing an important theoretical foundation for advancing the use of this subspecies as a model organism.

As research continues to advance, it has become evident that gut microbiota undergoes significant changes throughout an individual’s life. The imbalance in microbial composition is associated with age [49]. This study demonstrates that the gut microbiota of *M. mulatta brevicaudus* exhibits distinct differences and similarities across various age groups. The varying levels of Beta diversity and microbial abundance indicate that the gut microbiota of juvenile macaques is not yet stable and exhibits relatively low diversity. In contrast, as macaques age, their gut microbial communities become increasingly complex and stable, showing higher diversity; however, in old age, diversity tends to decrease again. These findings are consistent with studies conducted in humans [50,51]. The results reveal that *Segatella copri* is the most abundant species across different age groups. *S. copri* is part of the *S. copri* complex, which includes 13 distinct species [52]. This species is commonly found in the human gut and is estimated to have been colonizing humans for hundreds of thousands of years, serving as a common member of the human gut microbiome associated with health and disease states [53,54]. The presence of *Segatella copri* has been linked to the onset of various inflammatory diseases, such as rheumatoid arthritis [55,56]. Among the significantly different microbial species identified through random forest analysis, *Bifidobacterium adolescentis* was notably higher during the juvenile period compared to other stages. Related studies indicate that this species acts as a guardian of gut health in younger individuals, playing a crucial role in maintaining intestinal health and potentially possessing the ability to improve gut aging and delay overall aging in humans [57]. Additionally, research has shown that heat-inactivated *B. adolescentis* is a symbiotic bacterium primarily found in the intestines of younger individuals and may enhance the lifespan of multiple species [58]. Furthermore, studies indicate that *Desulfovibrio piger* varies with age in different populations, particularly exhibiting higher abundance in long-lived individuals [59,60]. Compared to existing research, the findings of this study suggest that the key microorganisms identified in *Macaca mulatta brevicaudus* are functionally convergent with human age, indicating that the gut microbiota of this macaque subspecies shares significant similarities with that of humans [47].

Sexual differences can significantly influence the composition and functionality of the gut microbiome, with these variations potentially related to a range of disease, physiological, immune, and behavioral factors [61,62,63,64,65]. This study found that the abundance of *Alistipes senegalensis*, a specific microbial species, was significantly lower in the feces of male macaques compared to females. Additionally, LEfSe analysis identified this species as one of the key contributors to the intergroup differences. *Alistipes senegalensis* is one of the 13 species within the *Alistipes* genus and is classified as an anaerobic, Gram-negative rod [66]. The study also revealed that several species within the *Parabacteroides* genus (including *Parabacteroides merdae*, *Parabacteroides johnsonii*, and *Parabacteroides sp. CT06*) exhibited significantly lower abundance in male macaques compared to females. Notably, *Parabacteroides* has been identified as a core member of the human gut microbiome [67], with detection rates exceeding 90% in a database of hundreds of thousands of human gut samples [68]. This genus may influence host health through various mechanisms, including modulating the immune system, which can play a dual role in autoimmune diseases [69,70], and regulating host metabolism to combat obesity and affect cardiovascular health [71]. The higher abundance of different microbial communities in female macaques may be attributed to the influence of female hormones. Estrogens could provide a more favorable growth environment for these species [72] and have the capacity to alter gut PH and nutrient availability [73,74], indirectly impacting the abundance of microbial communities.

The impact of different physiological states on the microbiome is closely related, particularly during infancy, which is a critical phase for rapid gut microbiota establishment. Changes in gut microbial populations during this period directly affect the normal growth and development of infants and their long-term health; therefore, studying the gut microbiota of mother macaques in various physiological states is vital for understanding its role in the development of their offspring [75,76,77]. This study found that *Alistipes indistinctus* was significantly less abundant during the lactation period compared to the non-lactation period. It was also identified as one of the species that most influenced the differences detected in LEfSe analysis between these two physiological states. Previous research suggests that *Alistipes indistinctus* plays a role as a driver in maintaining optimal levels of uric acid [78]. This species may affect the host’s various physiological states, particularly concerning gut health, immune responses, and metabolism [79,80]. Additionally, *Alistipes indistinctus* helps break down complex carbohydrates and proteins, enhancing nutrient absorption and improving the host’s nutritional status [81,82]. The study also indicates that the abundance of Capnocytophaga haemolytica is significantly higher during the lactation period compared to the non-lactation period. As a Gram-negative bacterium belonging to the *Capnocytophaga* genus, *C. haemolytica* is typically associated with the oral microbiome and may influence the balance within the oral microbial community, thereby promoting the development of various physiological states [83,84]. *C. haemolytica* can metabolize various carbohydrates and is involved in nutrient conversion in the oral cavity, contributing to the maintenance of microbial community balance [85,86]. In the lactating *M. mulatta brevicaudus*, the oral microbiome (including *Capnocytophaga haemolytica*) may be transmitted to infants via breast milk, thus influencing the establishment of the early infant microbiome. Additionally, by affecting the composition of the oral microbiota, it may regulate the immune status of the mother macaques, thereby indirectly influencing their traits and providing immunoprotection to their offspring.

This study conducted a comprehensive analysis of the gut microbiome gene functions in *M. mulatta brevicaudus* across different age groups, sexes, and physiological states (lactation vs. non-lactation). The findings revealed that middle-aged, male, and lactating macaques exhibited a higher number of genes in the major enriched functional pathways compared to other groups. This suggests that the gut microbiota in these categories has greater functional diversity. Functional enrichment analysis revealed significantly higher numbers of enriched pathways in middle-aged macaques compared to other age groups, with predominant enrichment in global and overview maps, carbohydrate metabolism, amino acid metabolism, and metabolism of cofactors. This pattern likely reflects optimized physiological homeostasis through gene enrichment in core metabolic pathways, highly coordinated energy and biomolecular metabolic networks, and maintenance of host health via multi-pathway collaboration of gut microbiota [87]. The observed higher pathway enrichment in male macaques may be attributed to their more extensive environmental exploration behaviors and greater energy demands. Studies demonstrate that male primates exhibit more active expression of basal metabolic genes due to their need for larger home ranges and territorial patrolling [88]. During lactation, the surge in energy requirements leads to comprehensive activation of core metabolic networks. Published evidence confirms that primates systematically upregulate fundamental metabolic pathways such as glycolysis during lactation [89,90,91]. By employing various methods to screen and analyze key microorganisms in the Hainan subspecies of macaques across different ages, sexes, and physiological states, the study aims to better understand the patterns of gut microbiota variation, community characteristics, and their functional impacts. This research further elucidates the mechanisms of interaction between hosts and their microbiota.

Our findings reveal conservation-relevant patterns in wild *M. mulatta brevicaudus*. The age-specific microbiome profile revealed that *Bifidobacterium adolescentis* reaches its peak abundance during juvenile development, functioning as a “guardian” of adolescent gut health. Notably, fluctuations in its fecal abundance may serve as a critical bioindicator for assessing habitat degradation in *M. mulatta brevicaudus* populations. Physiological states (lactation vs. non-lactation) associated microbiome can be understood as biomarkers for reproductive health surveillance.

## 5. Conclusions

In summary, the study identified the dominant microbial species across different age groups, which include *Segatella copri* and *Bifidobacterium adolescentis*. Among them, *Bifidobacterium adolescentis* may have the ability to improve intestinal health and delay aging. For different sexes, the dominant species were *Alistipes senegalensis* and *Parabacteroides.* The higher abundance of most bacterial communities in females may be due to estrogen providing a more favorable growth environment for these species. During lactation and non-lactation periods, the dominant species were found to be *Alistipes indistinctus* and *Capnocytophaga haemolytica*. The construction of *Alistipes indistinctus* during lactation can promote nutrient absorption and improve the nutritional status of the host. The key species identified through various groupings and methods provide a theoretical foundation for studying the metabolic functions of the gut microbiome in the *M. mulatta brevicaudus*. This research offers guidance for exploring the mechanisms and potential applications of gut microbiota influences across different ages, sexes, and physiological states.

## Figures and Tables

**Figure 1 biology-14-00980-f001:**
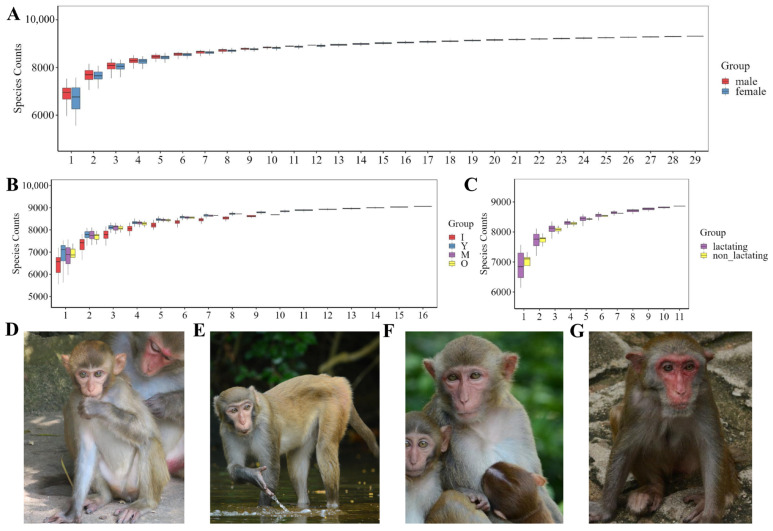
Boxplot of the dilution curve and pictures of different ages of *M. mulatta brevicaudus*. (**A**) Boxplot of dilution curves of different sexes. (**B**) Boxplot of dilution curves of different ages. (**C**) Boxplot of lactating and non-lactating macaques. (**D**) Infant. (**E**) Youth. (**F**) Middle. (**G**) Old. Note: The abscissa represents the number of samples, the ordinate indicates the number of species detected, and the color of the bins represents the grouping.

**Figure 2 biology-14-00980-f002:**
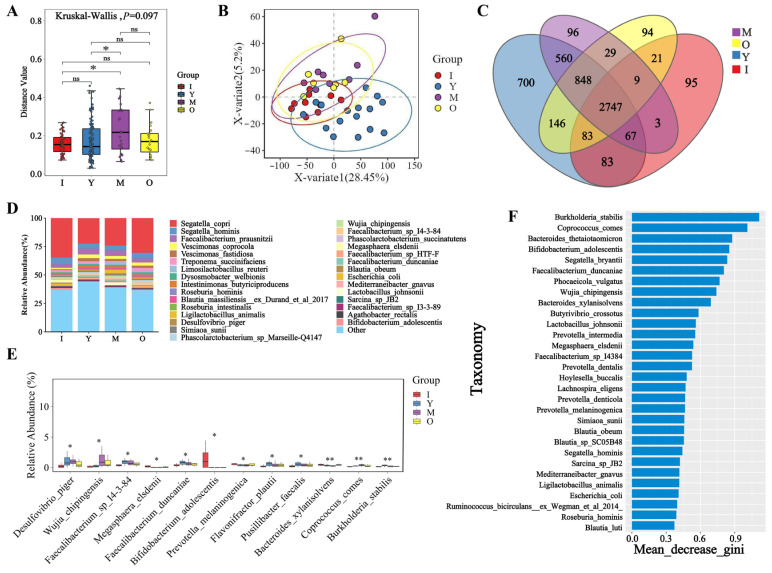
Species diversity and composition analysis across age groups in macaques. (**A**) Boxplot of species Beta diversity. (**B**) Scatter plot of PLS-DA analysis for different age groups. (**C**) Venn plots of species at the macaque species level at different ages. (**D**) Column stack map of species abundance. (**E**) Boxplot of species divergence. (**F**) Random forest ranking of the fecal microbial species level of macaques at different ages. Note: (**B**) The color of the point indicates the grouping of the sample. (**D**) The abscissa is the group, and the ordinate is the relative abundance of species. (**E**) The abscissa indicates the species with significant statistical test differences, and the ordinate indicates the relative abundance of species. "ns" indicates no significant difference between groups (*p* > 0.05), "*" indicates a significant difference between groups (*p* < 0.05), and "**" indicates a highly significant difference between groups (*p* < 0.01).

**Figure 3 biology-14-00980-f003:**
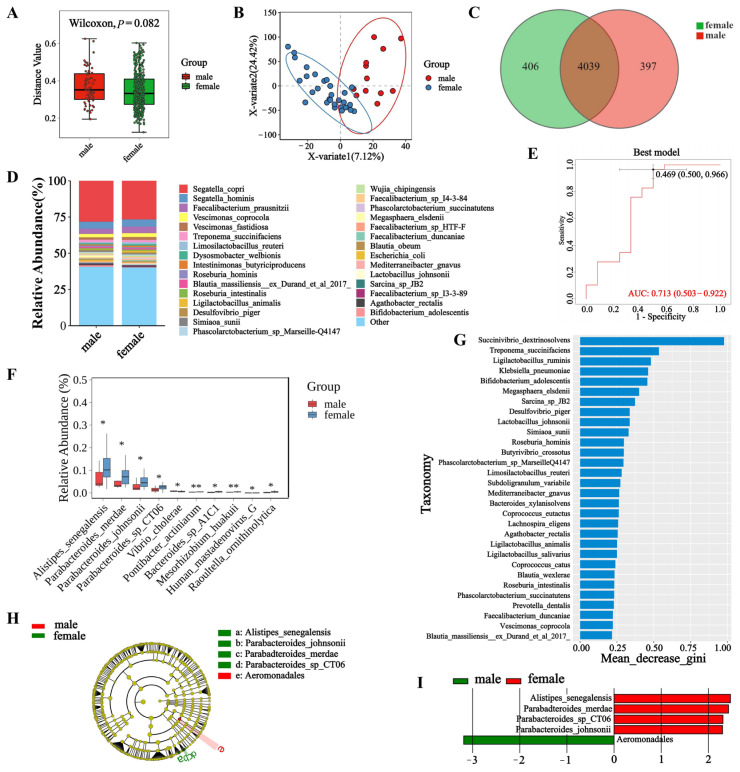
Comparative analysis of species diversity, composition, and biomarkers between male and female groups. (**A**) Boxplot of species Beta diversity. (**B**) Scatter plot of PLS-DA analysis between the two groups. (**C**) Venn diagram of species. (**D**) Column stack map of species abundance. (**E**) Different microbial ROC curves. (**F**) Boxplot of species divergence. (**G**) Random forest sorting. (**H**) LEfSe. (**I**) Bar chart of the distribution of LEfSe LDA values. Note: (**H**) A circle graph consists of multiple rings, where the inner ring is at a high taxonomic level, and the outer ring is at a low taxonomic level. If the classification with significant differences is not marked in the graph, the letter is marked in the graph. (**I**) The ordinate is the taxon with a difference, and the ordinate is the LDA value. "*" indicates a significant difference between groups (*p* < 0.05), and "**" indicates a highly significant difference between groups (*p* < 0.01).

**Figure 4 biology-14-00980-f004:**
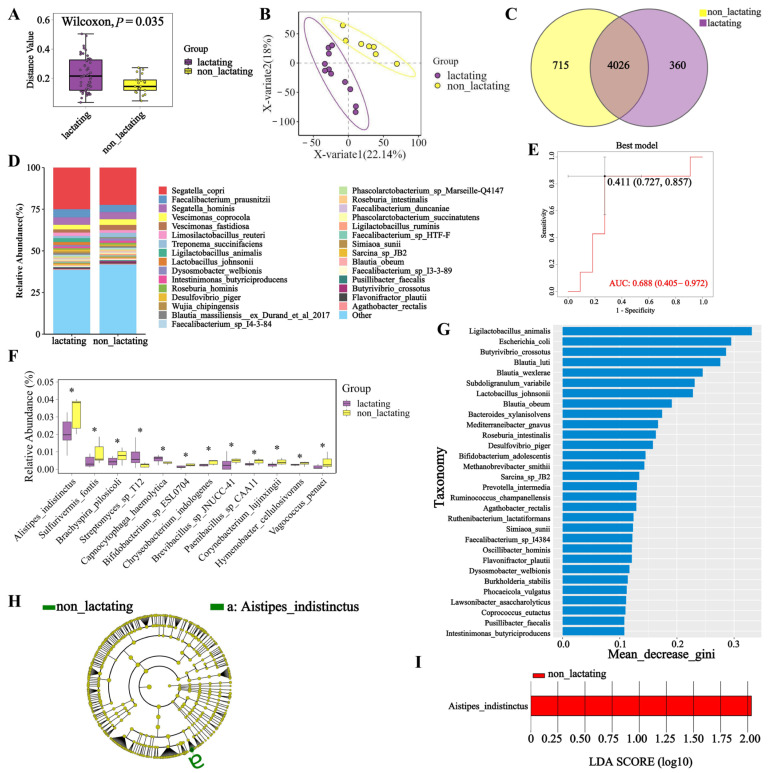
Comparative analysis of species diversity, composition, and biomarkers between lactation and non-lactation period groups. (**A**) Boxplot of species Beta diversity. (**B**) Scatter plot of PLS-DA analysis between the two groups. (**C**) Venn diagram of species. (**D**) Column stack map of species abundance; (**E**) Different microbial ROC curves. (**F**) Boxplot of species divergence. (**G**) Random forest sorting. (**H**) LEfSe. (**I**) Bar chart of the distribution of LEfSe LDA values. "*" indicates a significant difference between groups (*p* < 0.05).

**Figure 5 biology-14-00980-f005:**
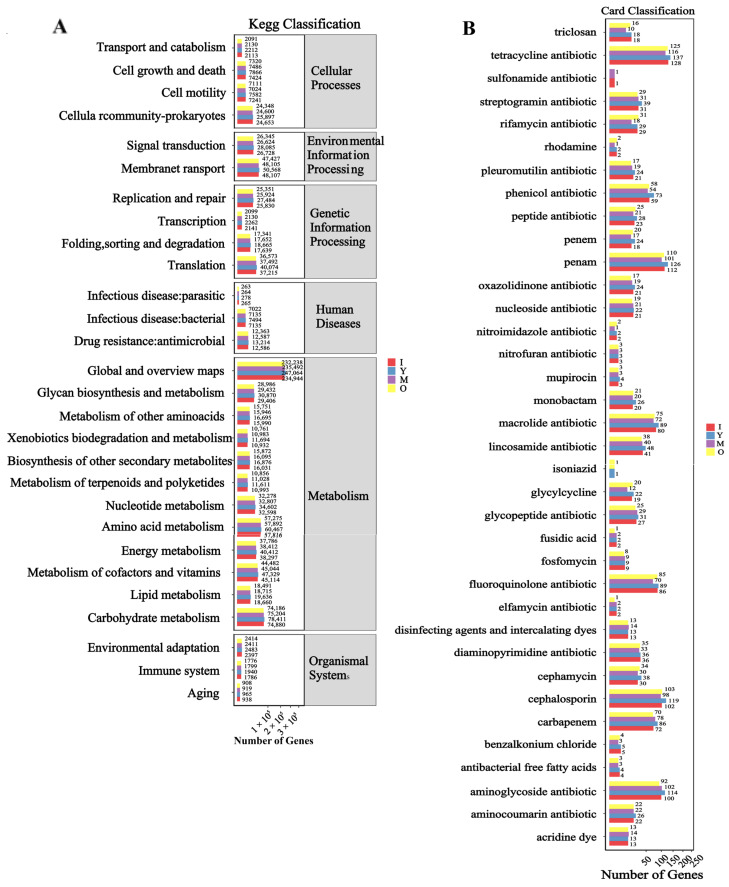
Bar graph of the statistics of functional genes for different ages. (**A**) Functional enrichment of KEGG. (**B**) Enrichment of antibiotic resistance genes. Note: The abscissa is the number of genes, and the ordinate is the functional classification.

**Figure 6 biology-14-00980-f006:**
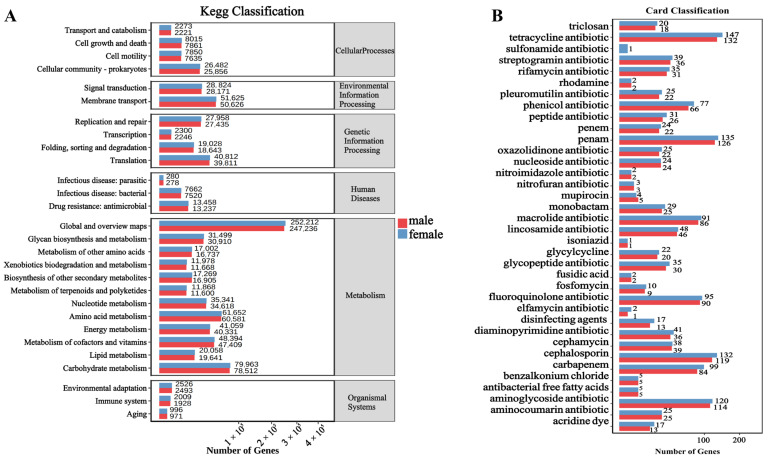
Bar graph of statistics of functional genes of different sex. (**A**) Functional enrichment of KEGG. (**B**) Enrichment of antibiotic resistance genes. Note: The abscissa is the number of genes, and the ordinate is the functional classification.

**Figure 7 biology-14-00980-f007:**
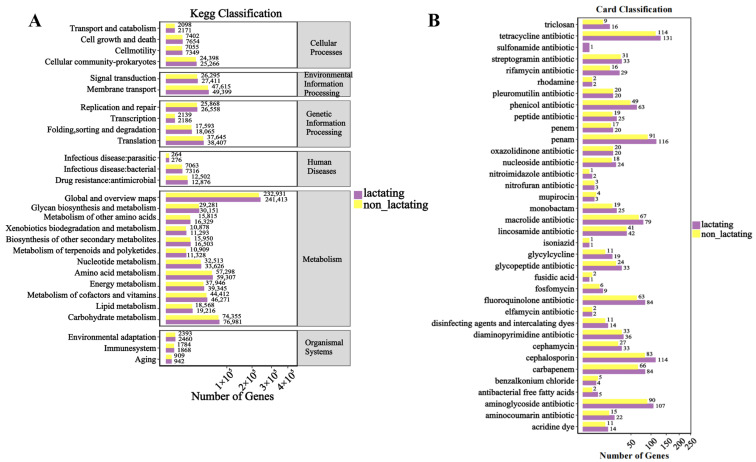
Statistical bar statistics of functional genes for different physiological states. (**A**) Functional enrichment of Kegg. (**B**) Enrichment of antibiotic resistance genes. Note: The abscissa is the number of genes, and the ordinate is the functional classification.

**Table 1 biology-14-00980-t001:** Group of *M. mulatta brevicaudus*.

Group		Number of Individuals in Different Age Groups					Total
		1–3 Years	4–6 Years	7–12 Years	13–17 Years	Over 17 Years Old	
Sex	Female	5	11	6	3	4	29
	Male	5	5	1	0	1	12
Lactation		0	2	6	2	1	11
Non-lactation		0	3	1	0	3	7
Weight/kg		2.32 ± 0.07	5.69 ± 0.61	7.65 ± 0.68	7.95 ± 0.63	7.97 ± 0.76	-

Note: The macaques were 1–3 years (infant, I), 4–6 years (young, Y), 7–12 years (middle, M), and 13 years (old, O).

## Data Availability

The raw sequence data reported in this paper have been deposited in the metagenomic Sequence Archive (Genomics, Proteomics & Bioinformatics 2021) in National Genomics Data Center (Nucleic Acids Res 2022), China National Center for Bioinformation/Beijing Institute of Genomics, Chinese Academy of Sciences that are publicly accessible at HTTPS: https://bigd.big.ac.cn/gsa/browse/CRA024773 (accessed on 29 July 2025) or FTP: ftp://download.big.ac.cn/gsa4/CRA024773 (accessed on 29 July 2025).

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
