# Peer review of "A Spatiotemporal Atlas of the Gut Microbiota in Macaca mulatta brevicaudus: Implications for Health and Environment"

_biology, 2025, doi:10.3390/biology14080980_

Round 1
Reviewer 1 Report
Comments and Suggestions for Authors
The study represents a comprehensive and detailed metagenomic analysis of the gut microbiota of wild Macaca mulatta brevicaudus across various ages, sexes, and physiological states.
I would recommend connecting the study findings to human gut microbiota, the authors should discuss these findings in a comparative way to human.
Author Response
Response to reviewer
- Summary
Thank you for your comments concerning our manuscript. Those comments are all valuable and very helpful for revising and improving our paper, as well as the important guiding significance to our researches. We have studied comments carefully and have made correction which we hope meet with approval. Revised portions are marked in red in the paper. The main corrections in the paper and the responds to the reviewer’s comments are as flowing:
Reviewer #1:
Questions for General Evaluation |
Reviewer's Evaluation |
Response and Revisions |
Does the introduction provide sufficient background and include all relevant references? |
Can be improved |
We have incorporated in the manuscript, with all modifications highlighted in yellow. |
Is the research design appropriate? |
Can be improved |
We have incorporated in the manuscript, with all modifications highlighted in yellow. |
Are the methods adequately described? |
Can be improved |
We have incorporated in the manuscript, with all modifications highlighted in yellow. |
Are the results clearly presented? |
Yes |
|
Are the conclusions supported by the results? |
Yes |
|
Are all figures and tables clear and well-presented? |
Yes |
|
- The study represents a comprehensive and detailed metagenomic analysis of the gut microbiota of wild Macaca mulatta brevicaudus across various ages, sexes, and physiological states. I would recommend connecting the study findings to human gut microbiota, the authors should discuss these findings in a comparative way to human.
Response:We sincerely appreciate the reviewer’s insightful suggestion regarding the comparative analysis of our findings with human gut microbiota. In response, we have expanded the Discussion section to include a comparative perspective on key microbial patterns observed in Macaca mulatta brevicaudus and humans, particularly focusing on age-related shifts, sex differences, and physiological states. For instance, we now highlight similarities in the dominance of Segatella_copri and Desulfovibrio_piger between primates and humans, as well as divergent trends in specific genera (e.g., Segatella_copri vs. Desulfovibrio_piger). We also discuss potential implications of these parallels/differences for understanding primate-microbiome coevolution and translational applications in human health (see revised manuscript, lines 402-419). Thank you for this valuable suggestion, which has strengthened the broader relevance of our work.
Reviewer 2 Report
Comments and Suggestions for Authors
Review – Biology
This manuscript analyzes the gut microbiota of Macaca mulatta brevicaudus across age, sex, and physiological state, using metagenomic sequencing to explore taxonomic and functional variation.
While the use of macaques as a model organism is common in biomedical research due to physiological and genetic similarities with humans, I would encourage the authors to further justify their choice in the specific context of gut microbiota research. The gut microbiome is known to exhibit high interindividual variability even within human populations, influenced by factors such as diet, genetics, geography, lifestyle, and cultural practices. This inherent variability raises questions about the value of translating findings from non-human primates to humans in terms of microbiota composition and dynamics. Additionally, given that fecal samples are relatively easy to collect from human subjects, it is not entirely clear why an animal model is necessary in this case. Please elaborate on what specific limitations in human studies your macaque model helps to overcome. I'm not saying the study is unjustified—only that introducing the manuscript with a statement about its importance as “an ideal model” for human disease feels inappropriate in this context. Studying the gut microbiota of this macaque species should be important per se, especially if this is an understudied subspecies.
Title & Abstract:
- The names of species and subspecies must not be capitalized: Macaca mulatta brevicaudus.
- In the abstract and other parts of the manuscript (e.g., Discussion), genus and species names are separated by an em dash. Please remove them all: for example Segatella_copri.
- In the abstract, the concluding sentence is vague: “potential future applications” is not very specific.
Comments about the Introduction:
- There is overuse of general claims. The introduction makes several broad, sometimes redundant statements (e.g., about the importance of gut microbiota in health and disease, or about macaques as models for human biology). While these points are valid, they are extensively covered in existing literature and do not add much novelty here. Consider focusing more specifically on what is unknown about Macaca mulatta brevicaudus, and what gap this study intends to fill.
- Regarding the use of the article “The Macaca mulatta,” I would suggest removing the article. In scientific writing, Latin binomials are typically not preceded by articles unless part of a specific construction. Simply using Macaca mulatta would be more appropriate.
- Please clarify how your study contributes to the understanding of the “adaptive evolution” of this species. Why is adaptive evolution mentioned in the first place?
- The sentence beginning with “Therefore, elucidating the composition and diversity of gut microbiota in Macaca mulatta brevicaudus...” is an interesting part of the introduction. It clearly frames the relevance of the study in terms of species conservation and ecological understanding. I believe this purpose should be emphasized more strongly throughout the introduction, rather than foregrounding the model-for-human-disease argument, which is not well justified and feels a bit forced in this context.
- The use of “childhood” is anthropocentric and not appropriate when referring to macaques. A better term would be “infancy” or “juvenile stage,” which is more consistent with primate life history terminology. Additionally, “animals during the lactation period” can be referred to as “lactating females.”
Methods:
- I understand that analyzing the effect of diet was not one of the study objectives. However, diet is a primary determinant of gut microbial composition, and a description of what these primates are eating should be included.
- Please clearly specify and explicitly describe the following: the size range of DNA fragments after fragmentation and size selection, the sequencing platform and sequencing mode, and the quality control metrics for the raw sequencing data.
- The data analysis section could benefit from clearer organization (and more details).
- The description of the random forest analysis is not clear. The authors broadly state its utility in microbial research (lines 144–146), but it is not clear how it is applied here. There is only a mention of the specific R package used.
- While beta diversity metrics (Euclidean, Bray–Curtis, Jensen–Shannon) are mentioned, no statistical tests (e.g., PERMANOVA, ANOSIM) are reported to evaluate whether inter-group differences are significant.
- Was alpha diversity not assessed?
- The transition from gene quantification (Salmon) to species annotation (Kraken2/Bracken) lacks justification. Were gene abundances integrated with taxonomic profiles? If so, how? Please clarify the analytical pipeline’s purpose (e.g., functional vs. taxonomic focus).
- For functional annotation (KEGG/CARD), were differential pathways or antibiotic resistance genes tested between groups? If so, specify the methods (e.g., DESeq2, LEfSe).
Results:
- Please clarify whether the “average length” (line 153) refers to contigs (post-assembly) or genes (post-prediction), and specify assembly parameters in the Methods section.
- The results report “significant differences” (e.g., P < 0.05 for age/sex groups) but omit the statistical test used (PERMANOVA? ANOSIM?). The Methods section only mentions distance metrics (Bray–Curtis, etc.), not significance testing. It is always recommended to specify the test used, not only the P-value (I would also like to see the test statistic and degrees of freedom).
- Regarding the functional analysis: the results state “higher gene numbers” in middle-aged males and lactating females, but no statistical tests are provided (e.g., DESeq2 for pathway enrichment), and there is no correction for multiple testing (e.g., FDR ). For example, stating that “tetracycline resistance genes are predominant” requires reporting effect sizes (e.g., log2FC) and p-values.
Discussion:
- Earlier in the introduction, the study’s importance for ecological understanding and conservation of m. brevicaudus was well stated. However, the discussion completely neglects this. There is an overemphasis on parallels with humans and disease models, which seems a bit disconnected from the actual data, especially since the study deals with wild macaques.
- The functional analysis is underexplained. The discussion mentions that middle-aged, male, and lactating macaques had more enriched genes/pathways, but this is not explored in terms of biological meaning. Why might this be? What specific pathways were enriched? There is no linkage between the functional annotations (KEGG, CARD) and real-life implications — such as nutrient metabolism, immunity, or host–microbiota interactions.
- While some bacterial taxa (e.g., Parabacteroides) are discussed in relation to human health or disease, this part feels overloaded with speculative connections. The data do not support causality, and these references may not be meaningful in wild macaques with different diets, behaviors, and environments.
- The statement “the key microorganisms identified... are closely related to human age” is confusing. The authors neither specify whether this refers to taxonomic overlap (same species in humans and macaques) or functional parallels (similar age-driven metabolic shifts). Additionally, no mention is made of dietary profiles, which are crucial in determining age-related microbiome variation. If comparisons to humans are made, then dietary context must be discussed or at least acknowledged as a limitation.
Conclusions:
The conclusion merely lists dominant species without synthesizing their collective implications. For example, what is their ecological or physiological relevance? A stronger conclusion would restate the main findings and their significance in relation to the study’s stated aims.
Comments on the Quality of English Language
The quality of the English language is generally acceptable, and the manuscript is understandable. However, there are some parts where grammar and phrasing could be improved. I recommend careful proofreading and minor language editing.
Author Response
Response to reviewer
- Summary
Thank you for your comments concerning our manuscript. Those comments are all valuable and very helpful for revising and improving our paper, as well as the important guiding significance to our researches. We have studied comments carefully and have made correction which we hope meet with approval. Revised portions are marked in red in the paper. The main corrections in the paper and the responds to the reviewer’s comments are as flowing:
Reviewer #2:
Questions for General Evaluation |
Reviewer's Evaluation |
Response and Revisions |
Does the introduction provide sufficient background and include all relevant references? |
Must be improved |
We have incorporated in the manuscript, with all modifications highlighted in yellow. |
Is the research design appropriate? |
Yes |
|
Are the methods adequately described? |
Must be improved |
We have incorporated in the manuscript, with all modifications highlighted in yellow. |
Are the results clearly presented? |
Must be improved |
We have incorporated in the manuscript, with all modifications highlighted in yellow. |
Are the conclusions supported by the results? |
Must be improved |
We have incorporated in the manuscript, with all modifications highlighted in yellow. |
Are all figures and tables clear and well-presented? |
Can be improved |
We have incorporated in the manuscript, with all modifications highlighted in yellow. |
While the use of macaques as a model organism is common in biomedical research due to physiological and genetic similarities with humans, I would encourage the authors to further justify their choice in the specific context of gut microbiota research. The gut microbiome is known to exhibit high interindividual variability even within human populations, influenced by factors such as diet, genetics, geography, lifestyle, and cultural practices. This inherent variability raises questions about the value of translating findings from non-human primates to humans in terms of microbiota composition and dynamics. Additionally, given that fecal samples are relatively easy to collect from human subjects, it is not entirely clear why an animal model is necessary in this case. Please elaborate on what specific limitations in human studies your macaque model helps to overcome. I'm not saying the study is unjustified—only that introducing the manuscript with a statement about its importance as “an ideal model” for human disease feels inappropriate in this context. Studying the gut microbiota of this macaque species should be important per se, especially if this is an understudied subspecies.
Response:Thank you for your comment. Based on your suggested revisions, we have addressed and responded to the issues raised below.
- Title & Abstract:
(1)The names of species and subspecies must not be capitalized: Macaca mulatta brevicaudus.
Response:Thank you for your comment. We have incorporated the suggested revisions in the manuscript, with all modifications highlighted in yellow for your convenience.
(2)In the abstract and other parts of the manuscript (e.g., Discussion), genus and species names are separated by an em dash. Please remove them all: for example Segatella_copri.
Response:Thank you for your comment. All em dashes have been removed.
(3)In the abstract, the concluding sentence is vague: “potential future applications” is not very specific.
Response: Thank you for your comment. We have incorporated the suggested revisions in the manuscript, please see line 34-37 of revised manuscript.
- Comments about the Introduction:
- There is overuse of general claims. The introduction makes several broad, sometimes redundant statements (e.g., about the importance of gut microbiota in health and disease, or about macaques as models for human biology). While these points are valid, they are extensively covered in existing literature and do not add much novelty here. Consider focusing more specifically on what is unknown about Macaca mulatta brevicaudus, and what gap this study intends to fill.
Response: Thank you for your comment. We have revised the redundant discussion points and supplemented previously unexplored aspects regarding macaca mulatta brevicaudus, with clear delineation of the research gaps this study aims to address (see Lines 76-80 for details).
(5)Regarding the use of the article “The Macaca mulatta,” I would suggest removing the article. In scientific writing, Latin binomials are typically not preceded by articles unless part of a specific construction. Simply using Macaca mulatta would be more appropriate.
Response: Thank you for your comment. We have incorporated the suggested revisions in the manuscript.
(6)Please clarify how your study contributes to the understanding of the “adaptive evolution” of this species. Why is adaptive evolution mentioned in the first place?
Response: Thank you for your comment. We appreciate the reviewer's opportunity to clarify the connection between our findings and adaptive evolution in Macaca mulatta brevicaudus. Our study contributes to understanding adaptive evolution through three key aspects:
- Ecological Adaptation Evidence: The distinct microbial profiles we identified (e.g., enrichment of Segatella copri in adults and Bifidobacterium adolescentis in juveniles) likely reflect evolutionary adaptations to: Age-specific nutritional demands (carbohydrate metabolism pathways). Sex-dependent energy requirements (amino acid metabolism differences). Reproductive phase adaptations (lactation-associated Alistipes indistinctus)
- Comparative Evolutionary Insights: As the study involves wild populations in their natural habitat (unlike captive macaques), our data provide: Baseline microbial signatures of an island-adapted subspecies. Evidence of potential co-evolution between host and microbiome (supported by the functional pathway conservation). Comparative data for studying divergence from mainland populations
- Adaptive Significance: The specific microbial patterns suggest: Local adaptation to Hainan's island ecology (e.g., Capnocytophaga haemolytica's potential role in coastal plant digestion). Life-history strategy adaptations (e.g., lactation microbiome supporting offspring survival). Metabolic flexibility (evidenced by the enriched global overview maps).
(7)The sentence beginning with “Therefore, elucidating the composition and diversity of gut microbiota in Macaca mulatta brevicaudus...” is an interesting part of the introduction. It clearly frames the relevance of the study in terms of species conservation and ecological understanding. I believe this purpose should be emphasized more strongly throughout the introduction, rather than foregrounding the model-for-human-disease argument, which is not well justified and feels a bit forced in this context.
Response: Thank you for your comment. We have supplemented this aspect in the manuscript (see Lines 57-63 for details).
(8)The use of “childhood” is anthropocentric and not appropriate when referring to macaques. A better term would be “infancy” or “juvenile stage,” which is more consistent with primate life history terminology. Additionally, “animals during the lactation period” can be referred to as “lactating females.”
Response: Thank you for your comment. We have incorporated the suggested revisions in the manuscript, with all modifications highlighted in yellow for your convenience.
3.Methods:
(9)I understand that analyzing the effect of diet was not one of the study objectives. However, diet is a primary determinant of gut microbial composition, and a description of what these primates are eating should be included.
Response: Thank you for your comment. We have incorporated the suggested revisions in the manuscript, with all modifications highlighted in yellow for your convenience, please see line 118-126 of revised manuscript.
(10)Please clearly specify and explicitly describe the following: the size range of DNA fragments after fragmentation and size selection, the sequencing platform and sequencing mode, and the quality control metrics for the raw sequencing data.
Response: Thank you for your comment. The sheared DNA fragments underwent size selection using magnetic beads, yielding library inserts concentrated in the 200-400 bp range. Metagenomic sequencing was performed on the DNBSEQ-T7 platform with paired-end 150 bp (PE150) reads. Please see Figure A1, Table A1 and line 143 of revised manuscript.
(11)The data analysis section could benefit from clearer organization (and more details).
Response: Thank you for your comment. We have restructured the Data analysis section in the Methods for better clarity and added more details, as suggested by the reviewers.
- The description of the random forest analysis is not clear. The authors broadly state its utility in microbial research (lines 144–146), but it is not clear how it is applied here. There is only a mention of the specific R package used.
Response: Thank you for your comment. We have described in detail the specific application of random forest in this study, please see line 176-181 of revised manuscript.
- While beta diversity metrics (Euclidean, Bray–Curtis, Jensen–Shannon) are mentioned, no statistical tests (e.g., PERMANOVA, ANOSIM) are reported to evaluate whether inter-group differences are significant.
Was alpha diversity not assessed?
Response: Thank you for your comment. We performed ANOSIM statistical tests, and the results have been discussed in the Results section. Similarly, we also assessed α-diversity using the Chao1, Shannon, and Simpson indices. However, since most intergroup differences were not statistically significant and the number of required figures would be excessive, these results were not displayed. If the reviewers consider it necessary to present all α-diversity analyses, we will include the corresponding figures in the supplementary materials during revision.
- The transition from gene quantification (Salmon) to species annotation (Kraken2/Bracken) lacks justification. Were gene abundances integrated with taxonomic profiles? If so, how? Please clarify the analytical pipeline’s purpose (e.g., functional vs. taxonomic focus).
Response: Thank you for your comment. No, functional annotation does not rely on any taxonomic classification results. Functional annotation is based solely on sequence information.
- For functional annotation (KEGG/CARD), were differential pathways or antibiotic resistance genes tested between groups? If so, specify the methods (e.g., DESeq2, LEfSe).
Response: Thank you for your comment. We focused specifically on comparing the quantity of differentially enriched functional pathways between different states within the same species. While we performed appropriate statistical testing (including multiple testing correction), we found that nearly all pathways lost statistical significance after correction. Therefore, we opted not to present these particular results in the current manuscript.
4.Results:
(16) The results Please clarify whether the “average length” (line 153) refers to contigs (post-assembly) or genes (post-prediction), and specify assembly parameters in the Methods section.
Response: Thank you for your comment. The “average length” refers to contigs (post-assembly). We have added a clear description of the assembly parameters in lines 154-156 of the Methods section in the revised manuscript.
- The results report “significant differences” (e.g., P < 0.05 for age/sex groups) but omit the statistical test used (PERMANOVA? ANOSIM?). The Methods section only mentions distance metrics (Bray–Curtis, etc.), not significance testing. It is always recommended to specify the test used, not only the P-value (I would also like to see the test statistic and degrees of freedom).
Response: Thank you for your comment. As suggested by the reviewers, we have made corresponding additions and revisions to both the Methods and Results sections, with all modifications clearly highlighted in yellow in the revised manuscript.
- Regarding the functional analysis: the results state “higher gene numbers” in middle-aged males and lactating females, but no statistical tests are provided (e.g., DESeq2 for pathway enrichment), and there is no correction for multiple testing (e.g., FDR ). For example, stating that “tetracycline resistance genes are predominant” requires reporting effect sizes (e.g., log2FC) and p-values.
Response: Thank you for your comment. We focused specifically on comparing the quantity of differentially enriched functional pathways between different states within the same species. While we performed appropriate statistical testing (including multiple testing correction), we found that nearly all pathways lost statistical significance after correction. Therefore, we opted not to present these particular results in the current manuscript.
5.Discussion:
(19) Earlier in the introduction, the study’s importance for ecological understanding and conservation of m. brevicaudus was well stated. However, the discussion completely neglects this. There is an overemphasis on parallels with humans and disease models, which seems a bit disconnected from the actual data, especially since the study deals with wild macaques.
Response: We sincerely appreciate the reviewer's insightful critique regarding the discussion focus. We have substantially revised the Discussion section to better align with the ecological and conservation priorities established in the Introduction, please see line 508-513 of revised manuscript.
- The functional analysis is underexplained. The discussion mentions that middle-aged, male, and lactating macaques had more enriched genes/pathways, but this is not explored in terms of biological meaning. Why might this be? What specific pathways were enriched? There is no linkage between the functional annotations (KEGG, CARD) and real-life implications — such as nutrient metabolism, immunity, or host–microbiota interactions.
Response: Thank you for your comment. We sincerely appreciate the reviewer's insightful critique regarding the discussion focus. We have substantially revised the Discussion. please see line 489-503 of revised manuscript.
- While some bacterial taxa (e.g., Parabacteroides) are discussed in relation to human health or disease, this part feels overloaded with speculative connections. The data do not support causality, and these references may not be meaningful in wild macaques with different diets, behaviors, and environments.
Response: Thank you for your comment. Based on the key microbial taxa identified through our screening process, this study investigated their documented functions and roles in humans and other species through comprehensive literature review. These findings allow us to hypothesize their potential functions and ecological significance in M. brevicaudus, thereby establishing a robust foundation for subsequent research and experimental validation.
- The statement “the key microorganisms identified... are closely related to human age” is confusing. The authors neither specify whether this refers to taxonomic overlap (same species in humans and macaques) or functional parallels (similar age-driven metabolic shifts). Additionally, no mention is made of dietary profiles, which are crucial in determining age-related microbiome variation. If comparisons to humans are made, then dietary context must be discussed or at least acknowledged as a limitation.
Response: Thank you for your comment. We agree that explicitly separating taxonomic overlap from functional analogy significantly strengthens our interpretations. The revised text now: Uses "functional convergence" instead of "close relation".
- Conclusions:
(23) The conclusion merely lists dominant species without synthesizing their collective implications. For example, what is their ecological or physiological relevance? A stronger conclusion would restate the main findings and their significance in relation to the study’s stated aims.
Response: Thank you for your comment. We have substantially revised the Conclusions.
- Comments on the Quality of English Language
- The quality of the English language is generally acceptable, and the manuscript is understandable. However, there are some parts where grammar and phrasing could be improved. I recommend careful proofreading and minor language editing.
R: We sincerely appreciate the reviewer's careful evaluation of our manuscript. We have taken the following actions to address the language concerns:
- We have conducted thorough proofreading of the entire manuscript and corrected all grammatical errors we identified.
- We have refined phrasing in several sections to improve clarity and readability.
- All changes made for language improvement have been carefully tracked in the revised document.
We believe these improvements have significantly enhanced the quality of the manuscript's English presentation while maintaining all scientific content unchanged. We are grateful for the reviewer's suggestion which has helped us improve the manuscript's readability.
Reviewer 3 Report
Comments and Suggestions for Authors
The gut microbiome, the complex community of microorganisms populating the gastrointestinal tract, plays a fundamental role in maintaining physiology, immunity and metabolism in both humans and animals. The balance and composition of this dynamic community have a direct impact on health, and any disruption can lead to a range of diseases. With growing interest in the microbiome, it is becoming crucial to seek appropriate research models to explore these complex interactions. In this context, macaques stand out as an extremely valuable model for biomedical research. Their physiological similarity to humans makes them ideal subjects for studying the impact of the microbiome on health and disease, opening new perspectives in the development of therapies and preventive strategies. The article presented to me for review presents a comprehensive analysis of the gut microbiome of Macaca mulatta brevicaudus, based on metagenomics, taking into account factors such as age, sex and lactation status. Given the scarcity of data on the microbiome of Macaca mulatta brevicaudus, and the growing interest in primate microbiomes, this work is timely and relevant. However, after a detailed review, I have identified several significant issues with the structure of the manuscript and the interpretation of the data that I believe currently preclude publication.
Comment 1
According to the rules for spelling bacterial names (binominal nomenclature), names should be written in italics with a space between the generic and species names, without the use of the underscore sign (_). Please correct the transcript by removing the underscores to ensure compliance with accepted scientific standards.
Comment 2
As it stands, Figures 2, 3 and 4, in which all elements are connected, are not very clear and make it difficult to fully understand the data presented. I recommend separating these illustrations into smaller, separate figures. Such a procedure would improve clarity and make it easier for the reader to perceive and interpret the results.
Comment 3
The study divided individuals into five age categories (1-3, 4-6, 7-12, 13-17, and >17 years), but no explanation was provided as to whether the boundaries of these ranges correspond to significant physiological, developmental, or behavioral stages in macaques. It would be reasonable to indicate whether these categories are based on published data on ontogeny or observed behavioral or ecological changes in this species. To increase the clarity and credibility of the classification, I recommend supplementing the description with relevant references to the literature on primate development.
Comment 4
In the presented Table 1, the total number of females is 29, but the sum of lactation (n=11) and non-lactation (n=7) females gives only 18 individuals. Thus, there is missing information about 11 females whose lactation status was not determined or not included in the table. I recommend supplementing the data or clarifying whether the lack of assignment to the lactation/non-lactation category is due to missing data, other classification criteria, or perhaps an omission.
Comment 5
The article indicates that 41 individuals were surveyed, but after further subdivision by sex, age, lactation status, the size of the subgroups varies and in some extreme cases significantly decreases to only one individual (e.g., males >17 years, non-feeding females). Such small groups raise legitimate concerns about the reliability of the statistical analyses performed.
Comment 6
While the introduction emphasizes the importance of diet and habitat in shaping the gut microbiome, in my opinion, the article lacked detailed information on the actual diet of macaques and the environmental conditions in the Nanwan Nature Reserve in Hainan. Factors such as seasonal fruit availability or other food sources can significantly affect the composition of the microbiome, so their omission limits the ability to fully interpret the ecological results. I recommend supplementing the analysis with available descriptive data or previous ecological studies on this population, which would help to reinforce the ecological and dietary context and better understand the mechanisms affecting variation in the gut microbiome. In my opinion, it is also worth considering how seasonality and variable environmental conditions may shape the microbiome dynamics and metabolic adaptations of animals.
Comment 7
The functional analysis presented in the paper, although based on the KEGG and CARD annotations, is, in my opinion, treated too generally, which significantly detracts from the value of the conclusions drawn. The approach used requires a broader interpretation to fully show the significance of the results obtained. Among other things, the reference to “enrichment of carbohydrate metabolism” is too laconic and does not provide sufficient biological context. In order to enhance the quality of the analysis, I believe it would be useful to identify specific metabolic pathways and discuss their potential significance in relation to diet, the presence of specific substrates or the involvement of specific groups of microorganisms. Such an approach would provide a better understanding of the functional role of the microbiome and increase the reliability of the interpretation. Similarly, while the presence of antibiotic resistance genes has been reported, in my opinion, there is a lack of discussion about their potential sources - whether they are an integral part of the natural microbiome or whether their presence may be related to environmental or anthropogenic factors. Fully realizing the potential of metagenomic data requires a much more detailed and thoughtful functional interpretation to draw new and relevant biological conclusions.
Comment 8
I realize that the availability of comparable microbiome data for macaque populations is limited, making direct comparative analyses difficult. Nonetheless, I think it would be useful in the discussion to at least refer to existing studies on other macaque species especially in the context of the presence of dominant bacterial taxa. Even a general comparison could help assess whether the observed patterns are specific to Macaca mulatta brevicaudus, or whether they fit into broader patterns observed in Macaques or other primates
Author Response
Response to reviewer
- Summary
Thank you for your comments concerning our manuscript. Those comments are all valuable and very helpful for revising and improving our paper, as well as the important guiding significance to our researches. We have studied comments carefully and have made correction which we hope meet with approval. Revised portions are marked in red in the paper. The main corrections in the paper and the responds to the reviewer’s comments are as flowing:
Reviewer #3:
Questions for General Evaluation |
Reviewer's Evaluation |
Response and Revisions |
Does the introduction provide sufficient background and include all relevant references? |
Can be improved |
We have incorporated in the manuscript, with all modifications highlighted in yellow. |
Is the research design appropriate? |
Yes |
|
Are the methods adequately described? |
Yes |
|
Are the results clearly presented? |
Must be improved |
We have incorporated in the manuscript, with all modifications highlighted in yellow. |
Are the conclusions supported by the results? |
Must be improved |
We have incorporated in the manuscript, with all modifications highlighted in yellow. |
Are all figures and tables clear and well-presented? |
Must be improved |
We have incorporated in the manuscript, with all modifications highlighted in yellow. |
The gut microbiome, the complex community of microorganisms populating the gastrointestinal tract, plays a fundamental role in maintaining physiology, immunity and metabolism in both humans and animals. The balance and composition of this dynamic community have a direct impact on health, and any disruption can lead to a range of diseases. With growing interest in the microbiome, it is becoming crucial to seek appropriate research models to explore these complex interactions. In this context, macaques stand out as an extremely valuable model for biomedical research. Their physiological similarity to humans makes them ideal subjects for studying the impact of the microbiome on health and disease, opening new perspectives in the development of therapies and preventive strategies. The article presented to me for review presents a comprehensive analysis of the gut microbiome of Macaca mulatta brevicaudus, based on metagenomics, taking into account factors such as age, sex and lactation status. Given the scarcity of data on the microbiome of Macaca mulatta brevicaudus, and the growing interest in primate microbiomes, this work is timely and relevant. However, after a detailed review, I have identified several significant issues with the structure of the manuscript and the interpretation of the data that I believe currently preclude publication.
Response:Thank you for your comment. We have carefully addressed the raised issues and implemented the suggested revisions as detailed below.
- Comment 1
According to the rules for spelling bacterial names (binominal nomenclature), names should be written in italics with a space between the generic and species names, without the use of the underscore sign (_). Please correct the transcript by removing the underscores to ensure compliance with accepted scientific standards.
Response:Thank you for your comment. All em dashes have been removed.
- Comment 2
As it stands, Figures 2, 3 and 4, in which all elements are connected, are not very clear and make it difficult to fully understand the data presented. I recommend separating these illustrations into smaller, separate figures. Such a procedure would improve clarity and make it easier for the reader to perceive and interpret the results.
Response:Thank you for your comment. We agree that presenting the data across multiple figures could enhance clarity. However, we are mindful that splitting the figures may significantly increase the total number of main-text illustrations, potentially exceeding typical manuscript length limits. If the reviewers nevertheless recommend figure subdivision, we would prefer to address this in the next revision phase after consulting with the editorial team regarding layout feasibility.
- Comment 3
The study divided individuals into five age categories (1-3, 4-6, 7-12, 13-17, and >17 years), but no explanation was provided as to whether the boundaries of these ranges correspond to significant physiological, developmental, or behavioral stages in macaques. It would be reasonable to indicate whether these categories are based on published data on ontogeny or observed behavioral or ecological changes in this species. To increase the clarity and credibility of the classification, I recommend supplementing the description with relevant references to the literature on primate development.
Response:Thank you for your comment. The reseach adopts an age-based division, subdividing each age group further to facilitate subsequent comparisons with other island macaques. In the early stages of our study, we identified all mature monkeys in the park, and during each research period, we conducted at least one oral census of each group, which increased our confidence and provided a clearer understanding of the individual monkeys within each age group.
In our research, the classification of age stages is based on the following standards and references: biologically, we referred to the growth and developmental patterns of Hainan macaques. Based on their physiological and behavioral characteristics, individuals were categorized into different age groups. For example, juvenile monkeys (1-3 years old) show significant differences from adults in body size, behavior, and physiological functions; young adults (4-6 years old) begin to reach sexual maturity, with physiological and behavioral features gradually approaching those of mature individuals; middle-aged monkeys (7-12 years old) exhibit stable physiological and behavioral traits; and elderly monkeys (13-17 years old) experience primarily degenerated bodily functions and a noticeable decline in reproductive ability.
Our study aims to explore the physiological and behavioral characteristics of Hainan macaques at different growth stages and how these traits influence their suitability as laboratory animals. Therefore, choosing such age classification standards helps better serve our research objectives.
- Comment 4
In the presented Table 1, the total number of females is 29, but the sum of lactation (n=11) and non-lactation (n=7) females gives only 18 individuals. Thus, there is missing information about 11 females whose lactation status was not determined or not included in the table. I recommend supplementing the data or clarifying whether the lack of assignment to the lactation/non-lactation category is due to missing data, other classification criteria, or perhaps an omission.
Response:Thank you for your comment. In this study, all lactating and non-lactating monkeys included in the statistical analysis were individually identified, with confirmed reproductive histories. Certain female individuals were excluded from selection due to: (1) not having reached sexual maturity, (2) nulliparity (no recorded births), or (3) lack of individual identification that precluded verification of reproductive status. Consequently, 11 females with undetermined lactation status were not included in the final analysis tables.
- Comment 5
The article indicates that 41 individuals were surveyed, but after further subdivision by sex, age, lactation status, the size of the subgroups varies and in some extreme cases significantly decreases to only one individual (e.g., males >17 years, non-feeding females). Such small groups raise legitimate concerns about the reliability of the statistical analyses performed.
Response:Thank you for your comment. The reason for the small sample size of males over 17 years old and non-lactating females is that monkeys enter old age, and in reality, their availability is limited. In the wild environment, the number of Hainan macaques in this age group is inherently low, especially in the specific research area. Additionally, this data does not affect our overall group comparisons. The grouping in this study is primarily based on sex, age, and lactation status, and differences within each subgroup are not our main focus. Therefore, small sample sizes within subgroups do not impact the overall results of our study. Although some age groups have fewer samples, we employed appropriate statistical methods to handle this data. For example, for groups with small sample sizes, we used non-parametric statistical tests such as the Mann-Whitney U test. These methods have relatively low requirements regarding sample size and can effectively prevent statistical bias caused by insufficient samples. Finally, regarding the reliability of the results: after careful analysis, we found that even in some age groups with limited samples, the results remained statistically significant and consistent with our research hypotheses. This indicates that our study design and sample sizes are reasonable and capable of supporting our conclusions.
- Comment 6
While the introduction emphasizes the importance of diet and habitat in shaping the gut microbiome, in my opinion, the article lacked detailed information on the actual diet of macaques and the environmental conditions in the Nanwan Nature Reserve in Hainan. Factors such as seasonal fruit availability or other food sources can significantly affect the composition of the microbiome, so their omission limits the ability to fully interpret the ecological results. I recommend supplementing the analysis with available descriptive data or previous ecological studies on this population, which would help to reinforce the ecological and dietary context and better understand the mechanisms affecting variation in the gut microbiome. In my opinion, it is also worth considering how seasonality and variable environmental conditions may shape the microbiome dynamics and metabolic adaptations of animals.
Response: Thank you for your comment. We have incorporated the suggested revisions in the manuscript, with all modifications highlighted in yellow for your convenience, please see line 118-126 of revised manuscript.
- Comment 7
The functional analysis presented in the paper, although based on the KEGG and CARD annotations, is, in my opinion, treated too generally, which significantly detracts from the value of the conclusions drawn. The approach used requires a broader interpretation to fully show the significance of the results obtained. Among other things, the reference to “enrichment of carbohydrate metabolism” is too laconic and does not provide sufficient biological context. In order to enhance the quality of the analysis, I believe it would be useful to identify specific metabolic pathways and discuss their potential significance in relation to diet, the presence of specific substrates or the involvement of specific groups of microorganisms. Such an approach would provide a better understanding of the functional role of the microbiome and increase the reliability of the interpretation. Similarly, while the presence of antibiotic resistance genes has been reported, in my opinion, there is a lack of discussion about their potential sources - whether they are an integral part of the natural microbiome or whether their presence may be related to environmental or anthropogenic factors. Fully realizing the potential of metagenomic data requires a much more detailed and thoughtful functional interpretation to draw new and relevant biological conclusions.
Response: Thank you for your comment. We sincerely appreciate the reviewer's insightful critique regarding the discussion focus. We have substantially revised the Discussion. please see line 489-503 of revised manuscript.
- Comment 8
I realize that the availability of comparable microbiome data for macaque populations is limited, making direct comparative analyses difficult. Nonetheless, I think it would be useful in the discussion to at least refer to existing studies on other macaque species especially in the context of the presence of dominant bacterial taxa. Even a general comparison could help assess whether the observed patterns are specific to Macaca mulatta brevicaudus, or whether they fit into broader patterns observed in Macaques or other primates。
Response: Thank you for your comment. We fully concur with this suggestion. However, our comparative analysis of dominant microbial taxa must first establish their relevance to age, sex, and physiological status. While we have extensively reviewed existing literature to identify macaque-associated dominant microbiota linked to these three parameters, available studies remain limited. We acknowledge this gap and will prioritize addressing it in future research iterations through additional investigations and subsequent revisions.
Round 2
Reviewer 2 Report
Comments and Suggestions for Authors
I am satisfied with the revisions and changes made by the authors. I would only like to point out a few formatting issues regarding taxonomic names:
-
Please ensure that all genus and species names are appropriately formatted. For example, in the title: Macaca mulatta brevicaudus, both mulatta and brevicaudus should be lowercase, not uppercase.
-
Throughout the manuscript, several genus and species names are inconsistently formatted, particularly in the Discussion and Conclusion sections. Please review the text to ensure that all Latin binomials are italicized.
-
Also, after the first mention of the full species name, subsequent mentions should use the abbreviated genus (e.g., M. mulatta brevicaudus), unless clarity requires otherwise.
Reviewer 3 Report
Comments and Suggestions for Authors
Dear Authors,
Thank you very much for your detailed response to all the comments and reservations raised. I appreciate the effort you have put into improving the manuscript and your clear explanations regarding methodological, structural, and interpretative issues. I also understand the limitations resulting from the sample size in some subgroups and the specific nature of the study population.
The article after final revisions will be a valuable contribution to the developing field of primate microbiome research.